# Do pay-for-performance schemes improve quality in community pharmacy? A mixed-methods study exploring stakeholder perspectives on implementation of the nationwide Pharmacy Quality Scheme (PQS) in England?

**Ellen Ingrid Schafheutle**[1]*, **Aidan Akira Moss**[2], **Ali Mawfek Khaled Hindi**[1],
**Jon Gibson**[3], **Emma Lovatt**[2], **Katie Robinson**[2], **Sally Jacobs**[1]

1 Centre for Pharmacy Workforce Studies, School of Health Sciences, Faculty of Biology Medicine and Health, The University of Manchester, Manchester, United Kingdom, 2 ICF Research and Evaluation Consulting Services, London, United Kingdom, 3 Division of Population Health, Health Services Research & Primary Care, School of Health Sciences, Faculty of Biology Medicine and Health, The University of Manchester, Manchester, United Kingdom

☙ These authors contributed equally to this work.
* ellen.schafheutle@manchester.ac.uk

## Abstract

### Main study objectives

To evaluate implementation and impact (at pharmacy and system level) of the pharmacy quality scheme (PQS), a pay-for-performance quality incentive scheme in community pharmacies in England since 2017.

### Methods

Mixed-methods evaluation. Three linked datasets for 2021/22 (n = 10,135) were analysed for impact of pharmacy size, type (independent, chain, supermarket), location, prescription volume, and region on PQS participation, domains completion and payments. Forty-one qualitative interviews conducted with pharmacists, employers and representative bodies explored views and experiences of PQS implementation and impact. Harrington et al's conceptual framework for evaluating community pharmacy pay-for-performance programmes guided qualitative data analysis.

### Results

Nearly all community pharmacies in England participated in PQS, with differences identified between chains (99% participation) and independents (16.5%), with income via PQS being an important motivator. Interviewees agreed with policy-makers about the purpose of the PQS being patient safety, patient experience, and clinical effectiveness. Beyond these core dimensions, consistency of service provision, sustainability, and wider system

**Data availability statement:** Data from this study cannot be shared publicly as consent was not given by participants to share the data publicly. To access the data upon approval contact cpws@manchester.ac.uk.

**Financial disclosure statement:** NHS England funded the 'Independent evaluation of the quality payments scheme (QPS) and pharmacy quality scheme (PQS) 2017-2021' in September 2022, award number C75805. Total funding: £177,540 (£75,214 of which to the University of Manchester). ICF was main contract holder, with Aidan Moss principal investigator (PI) at ICF. ICF subcontracted the University of Manchester, with Sally Jacobs as PI. Ellen Schafheutle (acting PI from November 2023), Jon Gibson and Ali Hindi were also recipients of the funding award as co-Is. The study team were independent of the funders, but they met regularly to discuss study focus and progress, to ensure accuracy. The funders did not influence data collection, analysis, interpretation, decision to publish, or preparation of the manuscript; they did have sight of the manuscript prior to submission, with an opportunity to comment on accuracy.

**Competing interests:** The authors have declared that no competing interests exist.

integration were considered important. While PQS was largely viewed as positively impacting pharmacy teams, clinical practice, and patient care, interviewees felt that increasing workloads across the sector made it challenging to focus on quality. They felt that there was a lack of feedback, that impacts were not always visible, and indeed frontline pharmacists were often not aware of published evidence of PQS impacts. Multiple sources of guidance lead to duplication and confusion. Particularly independent pharmacies found PQS workload burdensome and complex.

## Conclusion

The primary incentive for PQS engagement revolved around income stability for employers, with some positive impact achieved, but obstacles concerning resource implications and sustainability persist. Considering concerns about the viability of community pharmacy and the importance of increasing the scope of pharmaceutical services, these implementation challenges should lead policy-makers to question how best to incentivise quality.

## Introduction

Healthcare across the globe is experiencing significant pressures. In the developed world, these pressures stem from an ageing population with increasingly complex health needs, limited resources and growing healthcare workforce shortages. In addition to formulating cost-effective policies for healthcare delivery, there has been an increasing focus on improving and strengthening quality health and care systems to meet growing demand effectively. Public health and disease prevention have gained prominence, with many countries aiming to strengthen primary care provision, as gatekeepers into more specialist and costly healthcare services.

In England, primary care pressures have been exacerbated by significant workforce shortages in general practice. Over recent years, skill mix has been broadened, through the introduction of various roles, including advanced nurse practitioners, pharmacists, physiotherapists and physician associates [1]. There is now a recognition that a range of other providers may help ease pressures in primary care. One such provider are community pharmacies, which are often located in deprived areas and have the potential to address equity of access and health inequalities [2].

Community pharmacies in the United Kingdom (UK) are not alone internationally in offering increasing types of professional services [3], thus utilising the increasing knowledge of pharmacists and their teams to go beyond the more traditional roles of dispensing (medicines supply) and management of minor ailments. However, the UK has been at the forefront of these developments, with some differences across its devolved nations. In England, a significant change came in 2005 when, under the first Community Pharmacy Contractual Framework (CPCF) of its kind, a number of medicines (e.g., medicines use reviews [MURs]) and public health (e.g., drugs, alcohol) services were publicly (NHS) funded. This service offer was expanded in the 2019–24 CPCF, with services now increasingly aiming to relieve pressures in primary care, by making pharmacies patients' first port of call. One recent prominent example is Pharmacy First, under which community pharmacists can manage common conditions including infections, or manage and initiate contraception, using medicines otherwise only available on prescription.

It is imperative that these community pharmacy services are delivered to a high quality, meet patients' needs and trust, and integrate within patients' wider primary care pathways (avoiding duplication and confusion). However, under the CPCF, professional services were

introduced under a fee-per-service model, and evidence exists from countries such as Australia and Canada [4] that this can lead to activity being prioritised over quality. This was corroborated in a large national study which investigated factors associated with variation in MUR volumes [5,6], which found larger chains in particular pressurised pharmacists to meet MUR targets, thereby compromising quality [7].

NHS England (NHSE) as service commissioner, cannot (yet) pay community pharmacies like general practices, i.e. via an adjusted version of a per capita model. However, other ways of incentivising service quality may apply as equally to pharmacies as they do to general practice, such as regulation, professional standards, bonuses, awards, quality payments etc. [8]. Therefore, in 2017, NHSE, as the national commissioner of community pharmacy services, introduced its first attempt at incentivising quality in community pharmacy, by introducing the Quality Payments Scheme, which in 2019 was renamed the Pharmacy Quality Scheme (PQS). The PQS incentivises and rewards community pharmacy owners to achieve quality criteria in the three domains of clinical effectiveness, patient safety and patient experience, aligning with the NHS Five Year Forward View vision and Lord Darzi's [9] definition of quality care as clinically effective, personal and safe.

It is important to evaluate the effectiveness of PQS on quality in community pharmacy, and to gain insights into benefits and potential unintended consequences or negative impacts of the scheme. Only two US studies [10,11] have explored the design and impact of performance-based pharmacy payment models on community pharmacies. These studies showed that value-based pharmacy payment models have the potential to reduce healthcare costs and improve quality of care. However, these schemes can also pose challenges, including financial burden, frustration around staff and resource requirements for completion, and the difficulties of unclear quality criteria [11].

In England, the main incentive scheme in primary care/ general practice is the Quality and Outcomes Framework (QOF). QOF is a voluntary system to reward and incentivise quality via adherence to domain criteria related to patient experience, safety, and clinical effectiveness. The QOF scheme has changed considerably over time, with development of QOF indicators by the National Institute for Health and Care Excellence (NICE) beginning in 2009.The QOF provides most of the evidence that is currently available on quality incentive schemes [12–16]. Factors found to influence effectiveness of such schemes include: setting clear performance-reward links, prioritising group incentives, and taking incentive size into account [12]. However, assessing the long-term effects of incentive systems, matching incentives with quality goals, and preventing perverse incentives are challenges related to financial incentive schemes [12,15,16]. Healthcare professionals have also expressed concerns about diminishing autonomy resulting from narrowly focussing on meeting targets [15]. Systematic reviews looking at the QOF's effectiveness criticise its heavy reliance on process-based indicators without strong evidence of improvement in clinical outcomes [16]. The limitations of traditional top-down approaches like QOF suggest a need for patient-centred approaches and moving away from focusing on performance targets at the expense of wider conceptions of quality [8].

In light of the PQS being the first attempt in England to incentivise quality in community pharmacy, the very limited existing evidence on pharmacy quality schemes, and considering the expanding role of community pharmacies in an integrated primary care system which meets patients' needs effectively and efficiently, it is important to evaluate the implementation of the PQS.

## Aim

The aim of this study was to identify factors influencing participation in the PQS, explore participants' understanding of the purpose of the PQS and their experiences of taking part, and

determine whether they found the scheme beneficial, with a view to providing lessons learned for the implementation of similar quality initiative schemes.

## Methods

Primary data collection involved quantitative analysis of pharmacy achievement, costs and outcomes relating to the most recent year of PQS with available data (2021/22), in order to describe activity and outcomes data, if available. Qualitative interviews with 41 pharmacists, employers and representative bodies from across community pharmacy served to examine their understanding and views of the PQS, experiences of its implementation, and perceived impacts (intended or otherwise). When designing our study we drew on realist evaluation, which addresses the questions 'What works, for whom, in what circumstances and in what respects, and how?' [17], to understand what contributes to the success or otherwise of a particular programme.

### Quantitative data collection

PQS participation was voluntary, so descriptive quantitative analysis was used to analyse characteristics of community pharmacies involved in PQS, focusing on achievement data from 2021/22, which aligns with the latest PQS wave at the time of this study.

NHS England provided three datasets linked by postcode and organisational (ODS) code for analysis. The 2021/22 payments dataset (n = 10,135) included PQS domains completed, pharmacy band, points awarded, and payments received. The active pharmacies dataset (n = 11,134) listed pharmacies providing advanced services (which community pharmacies can choose to offer under the CPCF, as opposed to essential services which are offered by all) in 2021/22, identifying non-PQS participants. The characteristics dataset (n = 10,830) detailed pharmacy size, type (independent, chain, supermarket), location, prescription volume, and address.

Pharmacies with missing data were excluded, resulting in a final sample of 10,056 pharmacies (9,356 PQS participants, 700 non-participants). Three distance selling pharmacies (often online/ mail order pharmacies which cannot offer essential services face-to-face) were excluded to avoid skewing the analysis. Pharmacies in the payments dataset were classified as PQS participants, while active pharmacies not in the payments dataset were non-participants. Summary statistics were presented on characteristics such as operating hours, type, proximity to GPs and pharmacies, local deprivation, and NHS region.

### Qualitative data collection and analysis

The purpose of the qualitative interviews was to elicit the views and experiences of pharmacists and other stakeholders in relation to the PQS, gaining insights into how the PQS was understood and implemented. Sampling was purposive, and included three main groups: pharmacists, community pharmacy owners/ employers, their representative bodies (supported by Community Pharmacy England, previously known as the Pharmaceutical Services Negotiating Committee, who negotiate the CPCF and the PQS with NHSE), and system partners. Interviewees were recruited through a variety of methods, including advertising in Community Pharmacy England newsletters, social media, and snowball recruitment (e.g. several employers shared study information with their pharmacies).

Interview topic guides were informed by the programme theory developed by the research team for PQS, drawing on a realist evaluation approach. Different topic guides were developed for each participant group to ensure relevance, however, all topic guides covered stakeholders' perspectives on the following key areas: the interviewee's involvement with PQS; their

experience with PQS implementation; views on impacts of PQS; and any other reflections on the PQS and the wider policy context, including what could be done differently. University ethics committee approval was granted. Participants were provided with a participant information sheet and gave informed consent, which was recorded either in the form of a signed consent form, returned via email, or through verbal consent, recorded in an audio file stored separately from interview data (most opted for this option). Forty-one semi-structured interviews were conducted via Microsoft Teams between April and June 2023.

All interviews were audio-recorded, transcribed verbatim by a General Data Protection Regulation (GDPR) compliant transcribing company, and anonymised. Interview data were thematically analysed using a modified framework approach supported by NVivo qualitative analysis software. An initial *a priori* thematic framework based on the realist evaluation approach was developed and informed by the topic guides, and then further developed by the authors through familiarisation and regular discussion of identified themes. The coding frame was thus subject to ongoing and iterative development.

## Conceptual framework for evaluation of community pharmacy pay-for-performance programmes

This paper employs Harrington et al.'s conceptual framework for evaluating community pharmacy pay-for-performance (P4P) programmes, which incorporates elements from agency theory, psychology theory (extrinsic and intrinsic motivators), expectancy theory, and organisational theory [18]. The framework identifies four key domains: incentive, pharmacy, other influencing factors, and P4P programme measures, each with variables that may impact community pharmacy performance in a P4P scheme (Table 1). Empirical findings were mapped against the components of Harrington et al.'s conceptual framework. Using a combined iterative and theoretical approach to analysis ensured that Harrington's framework was applied in an exploratory manner to interpret the findings.

## Results

### Context: overview of PQS

To understand the implementation of the PQS, we first outline the financial scheme, specified quality domains, and present our analysis of participant characteristics. This groundwork provides the necessary context for subsequent qualitative work aimed at gaining a deeper understanding.

Table 1. Summary of Harrington et al.'s potential determinants of pharmacies' performance in a pay-for-performance programme [18].

| Domain | Variables |
|---|---|
| **Incentive** | • Financial incentive: amount, salience, recipient, payment schedule and type, timing, and purchaser size.<br>• Other characteristics: Control of quality measure, competing incentives, programme awareness, feedback, financial environment, nonfinancial incentives, and regulations. |
| **Pharmacy** | • Pharmacy management: Adaptability, entrepreneurial orientation, extrinsic and intrinsic motivators, management involvement, characteristics, and practice organization.<br>• Pharmacy operations: Structure, change management, processes, staffing, layout, resources, technology, and external support.<br>• Pharmacist motivators: Extrinsic - financial rewards, recognition, job conditions. Intrinsic - task accomplishment, job satisfaction, peer relationships. |
| **Other influencing factors** | • Organisational characteristics: Ownership, institutional layers, culture, change management, staffing, layout, resources, practice-setting type, technology, external support, and processes.<br>• Patient characteristics: Demographics, health status, medication adherence, engagement in care, socio-economic status, cultural factors. |
| **P4P measures** | • Quality measures: clinical relevance, perception, type, outcome type, performance thresholds, and number of measures. |

Introduced in 2017 (renamed from the Quality Payments Scheme [QPS] in 2019), the PQS uses annual financial incentives to improve provider performance. It is funded by reallocating part of the CPCF's global sum, with investments of £75m annually until 2022/23, reducing to £45m in 2023/24.

The PQS offers payments to community pharmacies for meeting annual quality criteria across various domains. Pharmacies must meet "gateway criteria" to be eligible for payments, which have evolved over time, including COVID-19 requirements in 2020–21. Payments increase with the number of completed domains and are now influenced by pharmacy banding, based on prescription volume.

Quality criteria are organised into domain areas (e.g. risk management, patient safety, safeguarding) and indicators within domain areas (e.g. conducting Centre for Pharmacy Postgraduate Education sepsis training and assessment). Each domain attracts a different number of points. Since 2021/22, pharmacies must complete all criteria within a domain in order to receive the points for that PQS domain. The split of points between overall domains is presented in Fig 1.

Fig 1 also shows how the PQS criteria have changed substantially over time. An early focus on domains covering digital elements, as well as risk management and patient safety, evolved gradually towards domains covering medicines safety audits, infection control, and antimicrobial resistance (AMR), as well as continued focus on asthma and respiratory domains. A Primary Care Network (PCN)-specific domain, encouraging the establishment of PCN leads was introduced in 2019/20 but withdrawn in 2022–23. PCN leads were responsible for providing strategic leadership, coordination, and implementation of healthcare initiatives within PCNs to improve integrated primary and community care services.

In order to support pharmacy contractors' completion of PQS criteria and domains, NHS England published detailed annual PQS guidance. Community Pharmacy England also issued

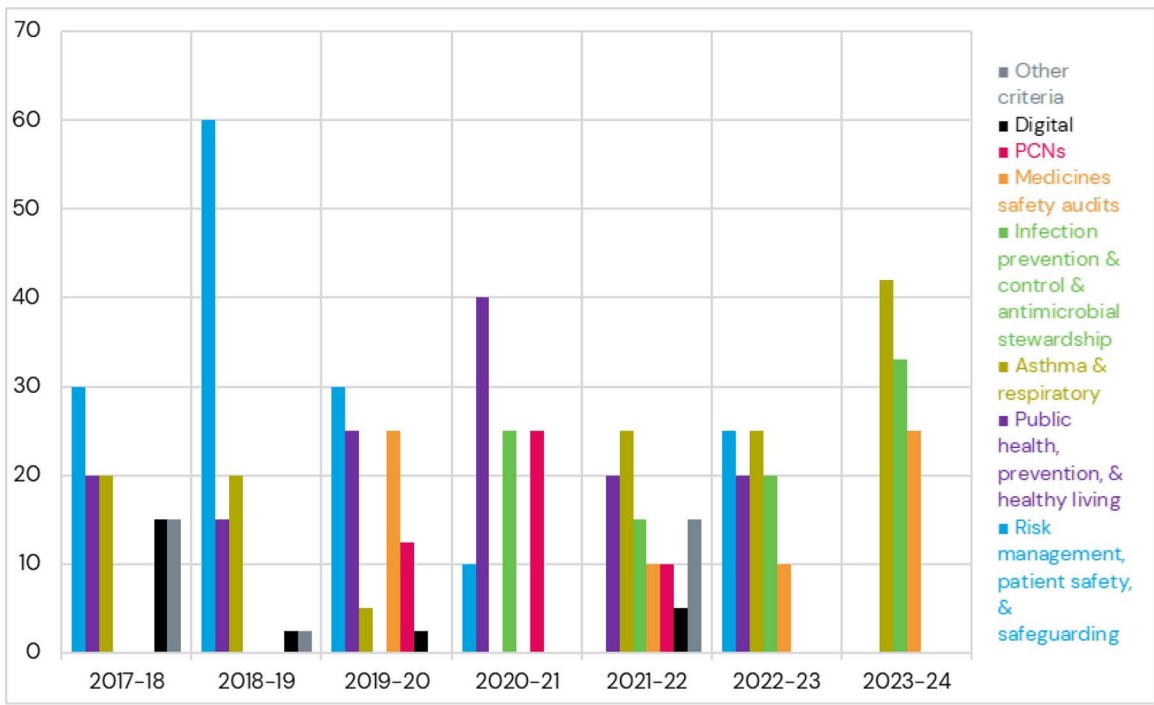

**Fig 1. Split of points per quality criteria domain per year (out of 100 total points).**

detailed PQS guidance every year (until 2023/24). In addition, head offices of larger multiples broke this guidance down further into smaller more focussed tasks with specific deadlines.

## PQS participation

Table 2 shows that pharmacies dispensing fewer than 8,000 items per month were less likely to participate in PQS (n = 475/6,059; 7.84%) than pharmacies dispensing more items (n = 225/3,997; 5.63% - p < 0.001). Independent pharmacies (those with 1–5 branches) were most likely not to participate in PQS (n = 656/3,968; 16.53%), whereas only 0.50% of large multiples (100 + branches) did not participate (n = 20/4,006 – p < 0.001). Less clustered pharmacies (with fewer other pharmacies within a 1km radius) were also significantly less likely to participate than those which were more clustered.

PQS payments are based on allocated points for each PQS domain and pharmacy banding (introduced in 2021/22), the latter being based on the pharmacy's number of items dispensed annually. Banding thresholds are displayed in Table 3, which also shows bands (2–6) by pharmacy type and region. Pharmacies with a greater total prescription item volume are assigned more points and consequently receive a higher payment.

**Table 2. PQS participation – Descriptive statistics.**

| Characteristics | Total | PQS non-participation | PQS participation | p-value |
|---|---|---|---|---|
| | N = 10,056 | N = 700 | N = 9,356 | |
| Items per month | | | | <0.001 |
| <8000 items per month | 6,059 | 475 (7.84%) | 5,584 (92.16%) | |
| 8000 + items per month | 3,997 | 225 (5.63%) | 3,772 (94.37%) | |
| 100 Hour Pharmacy | | | | <0.001 |
| No | 9,020 | 584 (6.47%) | 8,436 (93.53%) | |
| Yes | 1,036 | 116 (11.20%) | 920 (88.80%) | |
| Pharmacy Type (size of parent company) | | | | <0.001 |
| Independent (0–5 branches) | 3,968 | 656 (16.53%) | 3,312 (83.47%) | |
| Small/medium chain (6–99 branches) | 1,315 | 19 (1.44%) | 1,296 (98.56%) | |
| Multiple (100 + branches) | 4,006 | 20 (0.50%) | 3,986 (99.50%) | |
| Supermarket | 767 | 5 (0.65%) | 762 (99.35%) | |
| Average distance per item to GP practice (km) | 19.78 (10.22) | 20.22 (10.20) | 19.74 (10.22) | 0.24 |
| Number other pharmacies within: | | | | |
| 1km | 2.17 (3.73) | 2.23 (3.91) | 2.16 (3.71) | 0.68 |
| 5km | 3.03 (1.90) | 3.48 (1.91) | 2.99 (1.90) | <0.001 |
| Local population age 70 plus | 4.76 (0.88) | 4.78 (0.87) | 4.76 (0.88) | 0.42 |
| Deprivation of local population (IMD items) | 1453.64 (1618.94) | 1747.44 (2757.98) | 1431.66 (1497.26) | <0.001 |
| NHS Region | | | | <0.001 |
| East of England | 1,066 | 72 (6.75%) | 994 (93.25%) | |
| London | 1,642 | 243 (14.80%) | 1,399 (85.20%) | |
| Midlands | 1,945 | 133 (6.84%) | 1,812 (93.16%) | |
| North East and Yorkshire | 1,645 | 84 (5.11%) | 1,561 (94.89%) | |
| North West | 1,455 | 59 (4.05%) | 1,396 (95.95%) | |
| South East | 1,385 | 85 (6.14%) | 1,300 (93.86%) | |
| South West | 918 | 24 (2.61%) | 894 (97.39%) | |

Note: Continuous variables presented as mean and standard deviation in brackets. Binary or categorical variables presented as percentages in each category.

**Table 3. Descriptive statistics – banding.**

| | Total | Band 2 | Band 3 | Band 4 | Band 5 | Band 6 |
|---|---|---|---|---|---|---|
| | N = 9,358 | N = 276 | N = 1,686 | N = 6,454 | N = 823 | N = 119 |
| **Annual items dispensed for Banding category threshold** | | 1,201–30,000 | 30,001–60,000 | 60,001–150,000 | 150,001–230,000 | 230,001 + |
| **Pharmacy type (size of parent company)** | | | | | | |
| Independent (0–5 branches) | 3,312 (35.40%) | 59 (21.38%) | 544 (32.28%) | 2,254 (34.93%) | 376 (45.69%) | 79 (66.39%) |
| Small/medium chain (6–99) | 1,296 (13.85%) | 25 (9.06%) | 163 (9.67%) | 950 (14.72%) | 141 (17.13%) | 17 (14.29%) |
| Multiple (100 + branches) | 3,986 (42.60%) | 146 (52.90%) | 689 (40.89%) | 2,830 (43.86%) | 298 (36.21%) | 23 (19.33%) |
| Supermarket | 762 (8.14%) | 46 (16.67%) | 289 (17.15%) | 419 (6.49%) | 8 (0.97%) | 0 (0.00%) |
| **NHS region** | | | | | | |
| East of England | 994 (10.62%) | 12 (4.35%) | 166 (9.85%) | 717 (11.11%) | 84 (10.21%) | 15 (12.61%) |
| London | 1,399 (14.95%) | 113 (40.94%) | 463 (27.48%) | 750 (11.62%) | 64 (7.78%) | 9 (7.56%) |
| Midlands | 1,812 (19.37%) | 52 (18.84%) | 339 (20.12%) | 1,234 (19.12%) | 160 (19.44%) | 27 (22.69%) |
| North East and Yorkshire | 1,561 (16.68%) | 21 (7.61%) | 167 (9.91%) | 1,149 (17.81%) | 192 (23.33%) | 32 (26.89%) |
| North West | 1,396 (14.92%) | 27 (9.78%) | 225 (13.35%) | 1,000 (15.50%) | 130 (15.80%) | 14 (11.76%) |
| South East | 1,300 (13.89%) | 36 (13.04%) | 217 (12.88%) | 934 (14.47%) | 99 (12.03%) | 14 (11.76%) |
| South West | 894 (9.56%) | 15 (5.43%) | 108 (6.41%) | 669 (10.37%) | 94 (11.42%) | 8 (6.72%) |

NB. There were no data for band 1 (0–1,200 annual items dispensed).

## Qualitative interview findings

Qualitative interviews (lasting between 25–60 minutes) were conducted with 18 community pharmacists (9 independents (1–5 branches), 8 large multiples, 1 distance selling), 11 employers (2 medium, 9 large multiples), five representative bodies, and six system partners (NHSE Integrated Care Board clinical leads for community pharmacy).

This section presents an in-depth analysis of key stakeholders' views and experiences of the PQS, thus offering complementary, explanatory insights into our quantitative findings. Three themes emerged which were mapped onto Harrington et al.'s framework to provide insights into potential determinants of pharmacies' performance in the PQS (Table 4).

## Describing quality in community pharmacy and purpose of the PQS

Harrington et al.'s framework emphasises the importance of participants in a P4P programme understanding the metrics for quality measurement and comprehending the programme's complexity. When describing quality in community pharmacy, our interviewees consistently noted the three core dimensions of patient safety, patient experience, and clinical (or care) effectiveness.

> *"It's the right patient, right place, right time in terms of giving them the medication or service or advice that they need, and getting it right first time and every time and being able to have a workforce that if somebody rings up and needs some advice we have the people who can provide that advice. We have the people who can direct people to where the advice needs to be found and we're doing it at value for the NHS as well"* [Community pharmacist, distance selling pharmacy]

Besides these three quality pillars, our interviewees further identified the PQS as aiming to ensure that quality would be *consistent* across every community pharmacy; and also

**Table 4. Potential determinants of pharmacies' performance in a pay-for-performance programme (Harrington et al.) [18].**

| Themes (key stakeholders' views and experiences of the PQS) | Harrington et al. conceptual model variables [18] |
| --- | --- |
| Context/overview of the PQS (See quantitative section) | • Incentive recipient: institutional layers of the incentive through which an incentive must pass through (pharmacist vs. manager vs. owner or organisation); principal agent (payer recipient) relationship<br>• Payment schedule: frequent, small payments versus single lump-sum payment<br>• Payment type: incentive vs. baseline payment bonus vs. normal payment withheld, financial versus nonfinancial incentive; reward vs. punishment |
| Describing quality in community pharmacy and purpose of the PQS | • Programme awareness: familiarity with quality measure definition and complexity, in order for target actors to understand the programme |
| Facilitators and barriers to engagement with the PQS | • Financial salience: incentive award amount relative to resource input necessary to achieve quality measure (revenue potential and impact on recipient's costs to meet quality measure)<br>• Purchaser size: whether purchaser size is large enough to create an impact on the pharmacy with an incentive programme<br>• Extrinsic motivators: financial, fringe benefits, perquisites, patient appreciation, avoiding paperwork/bureaucracy, nature of job hierarchy, recognition/status, working conditions, stress<br>• Adaptability: degree to which management can adapt to market conditions/surroundings<br>• Structure: pharmacy staffing, pharmacy layout, resource adequacy, practice-setting type, technology present in setting, external support/assistance<br>• Change management: pharmacy's approach to aligning pharmacy resources to meet a quality measure performance threshold |
| Impact/outcomes of the PQS | • Measure perception: expectancy, valence, instrumentality<br>• Measure type: process measure, outcome measure, perceived objectivity of measure |

The themes identified in this PQS evaluation were mapped to the relevant variables of the Harrington et al. framework.

*cost-effective and sustainable.* Consistency was commonly referred to as setting 'minimum' or baseline standards for all community pharmacies, something which was viewed as important for pharmacy, the wider healthcare sector, and particularly patients.

> *"The first thing I would say is quality of patient care and consistency … Having a reassurance of a baseline … [which is] reassurance for patients, and commissioners, and other primary care healthcare professionals about patient safety, and about outcomes, and about the value of services offered being consistently good, whether that be dispensing medicines, or providing advice, or doing blood pressure checks, or providing advice for self care of a minor self limiting condition. … at the very least, all the minimum standards are met and ideally more"* [System partner, community pharmacy clinical role, NHS organisation]

Sustainability was raised, including financial viability and cost effectiveness, and indeed broader environmental sustainability.

> *"I think it's important that what we do is cost effective, and then final one is sustainable, and I mean sustainability both from the environmental and risk aspects."* [Employer, large multiple]

Whilst there was agreement on describing quality domains in community pharmacy and the purpose of the PQS, interviewees suggested that this vision was not always clearly communicated, particularly how the annual PQS domains and criteria fit into the wider NHS vision of quality integrated services. Interviewees also perceived a potential mismatch between the seemingly short-term nature of the annual PQS completion cycle and how PQS criteria fit into a longer term PQS vision.

> *"I would strip back the PQS to really focus on areas that were going to be longer term sustainable in pharmacy and potentially actually provide quality outcomes rather than just*

*being maybe a data collection exercise … Because it doesn't necessarily feel like there is much explanation of why we're doing this"* [Employer, large multiple]

There were comments on the lack of understanding of the PQS. While NHSE guidance was seen as helpful, it was also viewed as complex and time-consuming. Community Pharmacy England's (until 2023/24) detailed guidance was valuable but sometimes seen as duplicative or confusing.

Head offices of multiple pharmacies developed (with considerable resource) their own systems, timelines and instructions for PQS submissions. While breaking down PQS guidance into 'bite-sized' tasks facilitated submissions, it may have led to pharmacies viewing tasks without engaging deeply with the criteria's rationale and relevance to quality and patient care.

### Facilitators and barriers to engagement with the PQS

Harrington et al.'s framework identified several extrinsic motivators that can influence performance in a P4P scheme, namely financial incentives and the size of the purchaser, which must be large enough to create an impact on the pharmacy with an incentive programme. As seen in our quantitative analysis, almost all pharmacies participated in the PQS, their main motivation being financial income. The financial motivation was often coupled with resentment around the PQS funds having been reallocated from the global pharmacy sum, expecting contractors to do more to obtain income they would have been otherwise entitled to before the PQS. There was a sense that this left contractors with no choice but to participate in the PQS, and there was a small but vocal number of interviewees who perceived PQS to be a time consuming 'tick box exercise'.

*"I understand where the PQS started from and what it should be delivering. But if I'm looking at it very practically, and being very selfish and just focusing on my business, then I'd say that a lot of it is just a waste of time. You've taken our money away, and you're giving it back, telling us to do, jump through a lot of hoops for it"* [Employer, large multiple]

Harrington et al.'s framework highlights the concept of financial salience, which in our study was prevalent. This concept evaluates the incentive award amount relative to the resources required to achieve quality measures (such as revenue potential and impact on costs). Some interviewees expressed concerns over the amount of time required to complete the PQS, and there were some who did not complete all PQS domains. They weighed up the complexity and cost of PQS completion vs. the financial and other benefits to both the business and patients.

Concerns were also voiced that completion of the PQS was a detractor from core/ required activity, thus risking patient safety rather than enhancing quality. Staff shortages in the sector were highlighted as a significant barrier to being able to effectively resource the PQS, which resulted in increased pressure and workload on already stretched teams.

*"I think it's been a real challenge. Again it goes down to the staffing crisis that we've had over the last 24 months. But we are really struggling for support staff as well, and we've got more team vacancies than we've ever had, and finding the time to do the numerous training, the number of audits has been a particular challenge".* [Employer, large multiple]

Whilst PQS training criteria were generally viewed as beneficial by pharmacists and employers, some independent pharmacists and managers found them logistically challenging to implement, particularly during the working day.

*"So, for instance, if I'm using the consultation room, a team member cannot be in there doing their training. Whilst I do my training in my free time, it's not economically viable, or necessarily desirable, for pharmacy teams to do their training in their own time, because if they have questions or they don't understand it, there's no-one there to help them".* [Community pharmacist, independent pharmacy]

Harrington et al.'s framework also emphasises the importance of a pharmacy's change management strategy, specifically how it aligns resources to meet quality measure thresholds. Due to staffing pressures/ shortages, we found that particularly in independent pharmacies but also some multiples, staff had to use their own time to complete PQS tasks, mainly in relation to training.

In contrast, most pharmacists at large multiple chains reported not needing to use their personal time to complete PQS training.

*"I know within our organisation pharmacists are given extra time to complete this training and we do get paid a few extra hours of work just to make sure we do complete that training."* [Pharmacist, large multiple]

Only about half of interviewed employers reported involving locums in PQS, explaining that they lacked authority to ask locums to do tasks beyond core services. As many pharmacies were reliant on locums, with one interviewee from a large multiple stating that 40% of their pharmacies were currently locum run, this may be problematic.

*"The locums aren't incentivised to support PQS, so I guess that is one of the challenges of PQS is that you're not getting to the whole pharmacist population, you're only getting to those who either own their own business or are managed by a multiple or an independent multiple and have a drive to complete it in order to make the claim. So the locum population particularly aren't engaged in PQS, which is a big missed opportunity".* [Employer, large multiple]

Interviewees found some PQS criteria easier to complete than others. PQS criteria which aligned with what was required for other purposes were viewed positively, for example patient safety reports, which also have to be completed for General Pharmaceutical Council (GPhC) inspections.

PQS criteria which were dependent on people or organisations external to a community pharmacy were viewed as particularly difficult to complete. One example was around a stipulated need to engage with local general practices or PCN leads, where those with already strong links and good working relationships found meeting this PQS criterion easier than those without:

*"I work in an area where I would like to think my GP practice is quite proactive (…) We liaise quite well, we have meetings quarterly, and we bring up topics that we need to help each other out with (…) they are running a tight ship, so nothing really slips through the net because, yes, with the PQS, a lot of it was checking on patients (…) So if the doctor surgery was really on point, and doing really well, generally speaking pharmacy, with regards to the PQS, would have less work to do."* [Community pharmacist, independent pharmacy]

*"You're relying on third parties outside of the sector to engage (…) PCN Clinical Directors didn't want to engage at all. They weren't interested in wanting to engage with us. So, it made it very, very difficult".* [Community pharmacist, independent pharmacy]

## Impact/outcomes of the PQS

Harrington et al.'s framework addresses expectancy, valence, and instrumentality under Vroom's expectancy theory of motivation [19]. Expectancy involves the belief that one's effort will lead to high performance, hence better patient outcomes. Many interviewees noted how they had little if any understanding of the impact of their PQS engagement, at both a branch as well as wider NHS and patient care level.

Instrumentality is the perceived likelihood that performance will lead to rewards. Pharmacists appeared much more on board with PQS completion when they could see a direct link with patient care. This direct link enhanced the perceived value of their actions (i.e. valency), increasing their motivation towards PQS completion.

> "If I think that the thing I am doing is going to help the patients I see and work with every day, you are much more likely to want to do it and do it to a very high standard and deliver it well. If it's an abstract task in which you cannot see the purpose of it, it is much more difficult to get the motivation to do it. And it's much less likely to be done to the highest standard." [Representative Body, large multiples]

Many pharmacists expressed an interest in finding out what impact their PQS participation had, and would have been interested in receiving (more) feedback. Such feedback on their participation's impact would enhance pharmacists' expectancy, instrumentality, and valence.

> "I think if they understood that something is being done, they're not just doing these tasks and it being a tick box, that something is then being done with the results of the audits to show what the sector is doing and to showcase that, then I think that would influence people a bit more to understand that they are making a difference". [Representative Body, Community Pharmacy]

> "There's an awful lot of data which [PQS] has generated and then that's not shared with us to say, "this is what has happened nationally. These [are the] areas that we need to focus on"." [Community pharmacist, Independent pharmacy]

Most interviewees appeared unaware that NHSE had published the findings from some of the national audits (such as those relating to NSAIDs, valproate) [20,21]. Those interviewees who had seen the findings described them as being very helpful and a valuable reflection of their own contributions.

> "A few times they've released some data from audits they've conducted. So, like the anti-inflammatory audit, and the sodium valproate audit they did, [...] and that was quite refreshing for pharmacies, because it would come out with summary data. So it would say, 10,000 pharmacies conducted the anti-inflammatory audit. 20% found patients who were doing things they shouldn't be doing, or 5% of patients were prevented from going into hospital. And in that aspect it was good. [...] So even though I was a small drop in the ocean, as a collective I've done my bit to help patients" [Community pharmacist, independent pharmacy]

A small number of pharmacists (often working in independent pharmacies) did not feel the publication of feedback and outcomes would benefit them, as they expressed that their only reason for completing the PQS was to receive funding.

When discussing the impacts of engaging with the PQS, however, many interviewees identified positive outcomes on patients, pharmacists, pharmacy teams, community pharmacies, and the wider healthcare system. The PQS was seen as improving the consistency of services

and providing an annual focus on quality-related work, thus raising the overall quality of community pharmacy provision and patient care.

> *"What PQS I think has done, and for me the quality, it is that, actually, we should all be doing all these things, and it's incentivised that change and that consistent change so that we get, that bar is raised, and it's raised for most of the contractors." [System partner, community pharmacy clinical role, NHS organisation]*

## Impact of the PQS on patients

Interviewees noted that PQS participation had benefited patients through better access to services in community pharmacy, that these services would be delivered consistently in different pharmacies, and that patients had gained an increased understanding of, and confidence in, community pharmacy. Some health benefits were also identified, partly linked to increased knowledge and skills of pharmacy teams gained through PQS participation, including: recognition of health problems (hypertension, sepsis, cancer, obesity), identification of wrongly prescribed medicines, support for people experiencing domestic abuse, antimicrobial stewardship activities, recognising and acting on the specific needs of certain patient groups (e.g. dementia), and access to safe vaccinations.

> *"As part of the cancer awareness campaign, we referred a number of people to GPs for red flag symptoms of cancer and had patients coming back and saying, thank you [for] forcing me to go to the doctor, I'm now in an urgent appointment for their oncologist or for the relevant team for what they were referred to. So although that was unpaid, it drove pride, I guess, in our pharmacy teams that they were able to do that and able to identify that". [Employer, large multiple]*

Some interviewees felt the PQS had limited patient benefit despite significant resources required for its completion. A few suggested it could negatively affect patient care by diverting focus from patients' needs.

> *"I think there's just too much expected of us […]. I think it's just missed opportunities. While we're to focus on PQS, we're taking our eye off the ball elsewhere and potentially missing that one patient, his blood pressure's going through the roof and doesn't know it." [Community pharmacist and manager, large multiple pharmacy]*

Interestingly, even interviewees who lacked understanding or engagement with PQS inadvertently highlighted its positive impact. For example, undertaking audits requiring patient interaction shifted the focus from medicine supply to patient consultation, improving clinical practice.

## PQS impact on staff

PQS impact on staff was mainly related to stipulated training requirements for pharmacists and the whole team, which increased their understanding of certain conditions and patient needs. Engaging with other PQS criteria such as audits was also viewed positively, as they incentivised interaction with patients (and the wider healthcare team), leading to confidence building and embedding gradual change (and quality) into daily practice.

> *"I think that making NMS [New Medicine Service] a gateway criteria for this has probably improved the use of NMS. I think that, going right back to the beginning, the gateway criteria*

*of having to use summary care records, I just thought was ridiculous. I thought, why do people have to be told that? But they clearly weren't using it before. So I think that that made a big difference to practice. I think that made a difference to people using summary care records, it made it much more like normal practice." [Community pharmacist and manager, independent pharmacy]*

The main negative impact of the PQS on pharmacy teams related to the additional workload created for an already stretched workforce. This was felt particularly acutely by independents.

*"I can't describe to you how much of a mammoth task it was this year for me to do this, like really, really stressful because there was just so much and it felt really overwhelming at the beginning." [Community pharmacist and manager, independent pharmacy]*

## Impact of the PQS on workplaces

Interviewees viewed the PQS as having a positive impact on workplaces by introducing new processes and systems and providing opportunities for benchmarking and reflection.

*"I think the safety reports as a superintendent make me really think about where we can improve systems within the pharmacy. I hate doing them, but I love looking at the results if that makes sense... But I actually think they're really valuable to the running of the pharmacy." [Employer, LPC role]*

*"I think you'd probably find that people will continue with what they know, which are the prescriptions and the volumes, which aren't going away, whereas with this [PQS] you're incentivising people to change the mindset and move towards improving what they do on a daily basis." [System partner, community pharmacy clinical role, NHS organisation]*

Some noted that PQS helped shift the focus from dispensing to a wider role for community pharmacy in professional service delivery. Furthermore, the PQS criteria relating to training provided justification for some larger organisations to make the business case to create in-work/ funded time for training to upskill those working in a pharmacy.

*"I think staff always appreciate training, so it's a benefit from that that actually for me, as someone who would have lots more training within a business, it becomes a very useful tool for me to go to the powers that be who are maybe watching the pennies and say, "look this training has to be done. We need to do it. How can we fit it in?" [System partner, community pharmacy clinical role, NHS organisation]*

Nevertheless, the additional workload created by PQS was a concern, especially for independent pharmacies.

## Impact of the PQS on the wider health system

Overall, there were fewer reflections surrounding the impact of the PQS on the wider health system, with some interviewees describing how the PQS had positively impacted better integration of community pharmacy. There was also a perception that other primary care professionals' and patients' awareness of community pharmacy's role and contribution to the system had increased.

> *"I think the Pharmacy Quality Scheme has been one of the tools that has helped integrate pharmacy more and demonstrate, maybe advertise, the value of community pharmacy."* [System partner, community pharmacy clinical role, NHS organisation]

Some employers and pharmacists noted how increased communication with other healthcare professionals had resulted in community pharmacy being better integrated into patient care.

> *"Things like audits that obviously you have to communicate with, and flu vaccines, communication like at a PCN or a GP practice level, I think, really helped those working relationships. People started to talk to each other because it was mandated in PQS for them to do so."* [System partner, community pharmacy clinical role, NHS organisation]

A small number of interviewees perceived there would be benefits of further aligning the PQS with equivalent quality schemes for GPs in order to ensure components of the healthcare system were working harmoniously. This would create a more unified approach to care provision and further increase the chances of achieving positive health outcomes.

> *"These are the things GPs are doing. These are the things pharmacy are doing. What is the common thing that goes into the quality scheme for the pharmacy, that goes into the quality scheme for the GP? I think it needs to be worked in that manner… Rather than doing fragmented health campaigns or fragmented anything, we know when it's done on a national or a larger scale you get better health outcomes."* [System partner, community pharmacy clinical role, NHS organisation]

## Discussion

Drawing on quantitative and qualitative data, this study explored stakeholder views on the implementation of the PQS, which was introduced in England in 2017 as the first attempt at a scheme incentivising quality, or practice change for quality improvement, in community pharmacy as part of an integrated primary care system. Harrington et al.'s framework [18] served as an interpretative framework for this study, offering a structured post-analysis approach to offer plausible explanations for emerging themes.

Findings from this study drew out important considerations for effective implementation of pay-for-performance schemes in community pharmacy. Firstly, the implementation of a pharmacy quality scheme needs to hinge on a clear definition of what quality looks like in order to determine success and offer clarity for participants in the programme. Quality in community pharmacy has not been defined, but most interviewees could identify the purpose of the PQS under the core dimensions of patient safety, patient experience, effectiveness [9], as well as consistency in service offer and delivery, and cost-effectiveness and sustainability [22]. Our evaluation showed that the PQS has successfully engaged most community pharmacies and was perceived to have led to several positive impacts for patients, pharmacy teams, and the wider sector. These PQS impacts had been achieved by ensuring a baseline standard of quality and safety was met across the community pharmacy sector, with NHSE aiming to sustain and slowly increase these standards, by some PQS criteria subsequently becoming part of minimum terms of service under the CPCF. The PQS further ensured that pharmacy teams addressed key priorities for the NHS, and that training underpinned key NHS and clinical/patient safety areas. The PQS further introduced more consistent systems and clinical processes to develop the role of the community pharmacy sector, which supports the introduction of new services, and facilitates the operational foundations for integration of community pharmacy into the wider primary healthcare system.

Despite the purpose of the PQS and its direct and indirect positive impacts on quality, this study identified a number of contextual and logistical challenges. These related particularly to communication and guidance relating to the PQS, and the workload involved with understanding, operationalising and submitting PQS returns in an already financially stretched and pressured community pharmacy sector experiencing an acute workforce shortage.

Whilst there was agreement on defining what quality in community pharmacy ought to be, and the purpose of the PQS, interviewees suggested that this vision was not always clearly communicated, in particular how the annual PQS domains and criteria fit into the wider NHS vision of quality integrated services. Two US studies, which explored performance-based pharmacy payment models, also found that unclear criteria created challenges similar to those faced by the PQS [10,11]. Whilst NHSE guidance was reported to be clear, it was detailed and involved considerable time to operationalise. Many multiples took the approach of creating their own guidance via their head offices, which broke submissions down into clear tasks with deadlines. For independents, however, operationalising PQS required significant time and resource at the level of just a few pharmacies [1–5], suggesting why a greater proportion of these chose not to engage, as shown in our quantitative analysis. Indeed, wider evidence (mainly relating to QOF) suggests that engagement diminishes if the rewards of incentive schemes do not reflect the resources required [23].

Financial reimbursement was a significant driver for most community pharmacies, and indeed the vast majority engaged fully with the PQS for this reason, whilst independents were significantly less likely to engage than multiples. Although many employers understood the wider benefits of taking part, most were motivated to participate in the PQS because the income was important to the viability of their businesses. Whilst financial drivers are known to be a key contributor to engagement with quality incentive schemes [10,11], the reallocation of funds for the PQS from the global sum, and unprecedented workload and workforce pressures in community pharmacy, meant that participation was less of a choice than a financial necessity for business viability.

Because NHSE, Community Pharmacy England and head offices all produced guidance, these different information sources led to duplication and confusion, and created additional workload for some. It may therefore be valuable for NHSE to engage with key stakeholders (Community Pharmacy England, multiples, e.g. via Company Chemists Association, National Pharmacy Association etc.) to co-produce a combination of a task-focussed toolkit, accompanied by more detailed contextual information and an extended, supportive communications campaign. Research evidence suggests that incentive schemes need to be easily comprehensible [11] and that their purpose should be clearly understood, and the link with a vision for quality across the profession and the sector be explicit [24].

It appeared that even when participants did not fully understand the purpose of the PQS domains and submissions, and in some cases were very critical of the ability of such a scheme to have any impact on quality, PQS still achieved a positive impact. NHSE has published evidence on this impact nationally following PQS audit initiatives such as NSAIDS [25], valproate, high risk medicines, and oral anticoagulant safety [26]. Nevertheless, while much of the data – at a national level – were available and accessible, awareness appeared low, particularly amongst frontline pharmacists, even where implementation of recommendations from these reports was part of subsequent PQS criteria.

Besides national impact analyses, our qualitative interviews offered numerous illustrations of how undertaking PQS tasks, such as audits or training, had impacted practice positively. Interviewees described becoming more reflective of their practice, and there were examples where training appeared to embed skills in the workplace. Demonstrating this impact was outside the scope of this evaluation, but we do know from existing research that when

learning is applied and embedded, the resultant practice change is more likely to be sustained [27]. Nevertheless, this positive PQS impact on individual and organisational practice was not always evident to those involved. Interviewees wished for feedback and some form of benchmarking for their own performance against that of others. The creation of a toolkit could be considered, which could guide pharmacy staff to undertake their own benchmarking, in itself a potentially positive process of reflection, learning and implementation. Furthermore, this process could be integrated with pharmacy professionals' GPhC revalidation requirements, and thus guide the relevance of PQS learning to continuing professional development and impact on the patients they serve.

## Conclusions

This paper employed Harrington et al.'s conceptual framework for evaluating community pharmacy pay-for-performance programmes [18], by applying this to the PQS, the first P4P quality incentive scheme implemented nationally in England. We identified the following key contributors to success: 1) the need for more awareness/ clarity of what the programme intends to achieve, 2) ensuring the incentives are of sufficient value (financially and otherwise) to be worthy of resources/ staff investment (considering pharmacy capabilities, and different pharmacy sizes/types), 3) communication of outcomes, including whether and how participation benefits. These insights can inform other quality incentive schemes (particularly for sectors where such schemes are less established, such as dentistry and optometry).

## Acknowledgments

We would like to thank NHS England and its partners for commissioning this independent evaluation. We would like to extend particular thanks to the Pharmacy Quality Scheme team (NHSE, Community Pharmacy England, and DHSC) for steering this study and for supporting engagement with key stakeholders. We are grateful to the NHSE analytics team, the NHS BSA, and the NHSE policy team for guidance and supporting data access. We would also like to thank all the pharmacists across the sector and their representatives who took part in our interviews, and who generously gave their time to tell us about their experiences of the PQS and its impact; as well as the patient and organisational representatives who took part in key informant interviews and the stakeholder engagement workshop.

Finally, we would like to thank Selma Stearns at ICF for her role in the early stages of study design, data analysis and report writing.

## Author contributions

**Conceptualization:** Ellen Ingrid Schafheutle, Aidan Akira Moss, Ali Mawfek Khaled Hindi, Jon Gibson, Sally Jacobs.

**Data curation:** Aidan Akira Moss.

**Formal analysis:** Ellen Ingrid Schafheutle, Aidan Akira Moss, Jon Gibson, Emma Lovatt, Katie Robinson, Sally Jacobs.

**Funding acquisition:** Ellen Ingrid Schafheutle, Aidan Akira Moss, Ali Mawfek Khaled Hindi, Sally Jacobs.

**Investigation:** Ellen Ingrid Schafheutle, Aidan Akira Moss, Ali Mawfek Khaled Hindi, Jon Gibson, Emma Lovatt, Katie Robinson, Sally Jacobs.

**Methodology:** Ellen Ingrid Schafheutle, Aidan Akira Moss, Ali Mawfek Khaled Hindi, Jon Gibson, Sally Jacobs.

**Project administration:** Ellen Ingrid Schafheutle, Aidan Akira Moss, Emma Lovatt, Katie Robinson, Sally Jacobs.

**Resources:** Ellen Ingrid Schafheutle, Aidan Akira Moss, Sally Jacobs.

**Supervision:** Ellen Ingrid Schafheutle, Aidan Akira Moss, Sally Jacobs.

**Validation:** Aidan Akira Moss, Sally Jacobs.

**Visualization:** Aidan Akira Moss.

**Writing – original draft:** Ellen Ingrid Schafheutle.

**Writing – review & editing:** Ellen Ingrid Schafheutle, Aidan Akira Moss, Ali Mawfek Khaled Hindi, Jon Gibson, Emma Lovatt, Katie Robinson, Sally Jacobs.

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
