## [Decision Letter · Decision Letter 0]

20 Oct 2024

PONE-D-24-37890Do pay-for-performance schemes improve quality in community pharmacy? A mixed-methods study exploring stakeholder perspectives on implementation of the nationwide Pharmacy Quality Scheme (PQS) in England?PLOS ONE

Dear Dr. Schafheutle,

Thank you for submitting your manuscript to PLOS ONE. After careful consideration, we feel that it has merit but does not fully meet PLOS ONE’s publication criteria as it currently stands. Therefore, we invite you to submit a revised version of the manuscript that addresses the points raised during the review process.

We look forward to receiving your revised manuscript.

Kind regards,

Vijayaprakash Suppiah, PhD

Academic Editor

PLOS ONE

2. Please provide additional details regarding participant consent. In the ethics statement in the Methods and online submission information, please ensure that you have specified (1) whether consent was informed and (2) what type you obtained (for instance, written or verbal, and if verbal, how it was documented and witnessed).

“NHS England funded the 'Independent evaluation of the quality payments scheme (QPS) and pharmacy quality scheme (PQS) 2017-2021' in September 2022.  Total funding: £177,540 (£75,214 to the University of Manchester, the rest to ICF)”

4. In the online submission form, you indicated that your data is available only on request from a third party. Please note that your Data Availability Statement is currently missing the name of the third party contact or institution / contact details for the third party, such as an email address or a link to where data requests can be made. Please update your statement with the missing information.

Reviewers' comments:

Reviewer's Responses to Questions

**Comments to the Author**

1. Is the manuscript technically sound, and do the data support the conclusions?

Reviewer #1: Yes

2. Has the statistical analysis been performed appropriately and rigorously? 

Reviewer #1: Yes

3. Have the authors made all data underlying the findings in their manuscript fully available?

Reviewer #1: No

4. Is the manuscript presented in an intelligible fashion and written in standard English?

Reviewer #1: Yes

5. Review Comments to the Author

Reviewer #1: Thank you for the opportunity to review this manuscript. This is a very well written manuscript and worthy of publication.

I would like to make a few comments for the authors to consider:

• Line 120 What do the authors mean by advanced services?

• Line 124 For the international reader, please explain what is meant by ‘distance selling pharmacies’.

• Line 144 The information of the duration of interviews should be in the results section.

• Line 146 What is meant by GDPR?5

• Lines 168 The overview of PQS should be moved to the introduction as it does not appear to be a result of the current study

• What is meant by ‘banding’. This may not be clear for the international reader.

• Table 4 is confusing

• Line 313 This quote was already stated in Line 304

• Line 546 Suggest not to use ‘etc’.

• Congratulations on a well-written manuscript.

6. PLOS authors have the option to publish the peer review history of their article (what does this mean? ). If published, this will include your full peer review and any attached files.

**Do you want your identity to be public for this peer review?** For information about this choice, including consent withdrawal, please see our Privacy Policy .

Reviewer #1: **Yes: ** Petra Czarniak

---

## [Author Response · Author response to Decision Letter 1]

17 Dec 2024

Our response to each reviewer comment is detailed in the attached response table.

---

## [Editor Report · Decision Letter 1]

29 Jan 2025

Do pay-for-performance schemes improve quality in community pharmacy? A mixed-methods study exploring stakeholder perspectives on implementation of the nationwide Pharmacy Quality Scheme (PQS) in England?

PONE-D-24-37890R1

Dear Dr. Schafheutle,

We’re pleased to inform you that your manuscript has been judged scientifically suitable for publication and will be formally accepted for publication once it meets all outstanding technical requirements.

Kind regards,

Ilhem Berrou, PhD

Academic Editor

PLOS ONE
---

## [Editor Report · Acceptance letter]

PONE-D-24-37890R1

PLOS ONE

Dear Dr. Schafheutle,

I'm pleased to inform you that your manuscript has been deemed suitable for publication in PLOS ONE. Congratulations! Your manuscript is now being handed over to our production team.

Kind regards,

on behalf of

Dr. Ilhem Berrou

Academic Editor

PLOS ONE